# Migration as a Challenge to the Sustainability of Nordic Gender Equality Policies as Highlighted through the Lived Experiences of Eritrean Mothers Living in Denmark

**Natalie Joubert *, Janet Carter Anand** 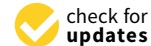 **and Tomi Mäki-Opas**

Department of Social Sciences, University of Eastern Finland, 70211 Kuopio, Finland; janet.anand@uef.fi (J.C.A.); tomi.maki-opas@uef.fi (T.M.-O.)

* Correspondence: njoubert@uef.fi

**Abstract:** This study focused on the complex process of adjustment and adaptation experienced by refugee parents from Eritrea who have settled in Aalborg, Denmark. Migration is a challenge to the sustainability of Nordic gender equality policies, in the face of cultural differences between refugees and host countries. This narrative study undertaken in the Eritrean community in Aalborg, Denmark took place against the background of cultural differences between the refugees and their host country, and Nordic gender equality policies. The study was done through the lens of parenting, to provide Eritrean refugee parents in Aalborg with the opportunity to share their lived experiences of settling in Denmark. The overarching aim of this study was to explore with Eritrean parents how they raise their children in a new country, as well as identifying both the challenges they face and the strengths which they bring to that role through their narratives. It aims to improve the understanding of what is significant to these parents during the process of their adaptation to a new environment. The role of refugees is well-established in their country of origin, but exposure to the Nordic Welfare Model which embraces women as being equal to men, is often problematic for Eritrean female refugees. Increasing cross-cultural knowledge in Denmark, through becoming aware of the lived experiences of the refugees as parents is important, particularly for those involved in social services that engage with this community. The study focused on the nature of challenges faced by Eritrean mothers experienced whilst integrating into Danish society. A semi-structured approach was used to obtain and analyze the data that was collected through interpersonal, qualitative methods in a narrative paradigm. The methodology was informed by initial focus groups meetings. Face-to-face engagement with the parents, utilizing an Eritrean interpreter as an integral part of the research team, was used. This study has highlighted the importance of engaging directly with refugee communities within their existing structures with a willingness to understand their culture. This approach may sit outside traditional research settings and service provision norms, but it informs more targeted, culturally appropriate, and acceptable interventions, which will assist the refugee community to effectively integrate into Danish society. The questions raised indicate an urgent need to recognize the cultural differences between refugees and host countries, and for this purpose to obtain more in-depth studies addressing this poorly examined area.

**Keywords:** gender equality policy; Danish Welfare State; Eritrean women refugees; culture and parenting roles; explorative

## 1. Introduction

"Immigrant women do not have contact with the surrounding society. They do not know their rights, and they do not determine their own existence. The lack of gender equality for many immigrant women is part of the new inequality. It is important to create equal opportunities between men and women because this contributes to creating strong cohesion in society."

This quote in 2007 by a Danish parliamentarian illustrates current perceptions of perceived inequality experienced by refugee women [1]. This exemplifies what has been highlighted by a number of studies, emphasizing the difficulties in attitude that the Nordic welfare model has towards meeting the needs of those with a multicultural background [2–5]. Parenting is an area that can be affected by this attitude. In theory, public services are designed to benefit all genders and diversified groups of people but emerging research has highlighted how certain service providers can be judgmental towards some cultural groups by labeling them as being patriarchal [6]. Ebot [5] in addressing this argues that Nordic family policies focus on the dual/earner/dual-carer model and do not address the cultural and religious values of ethnic minorities. In a study conducted in Finland with African parents, he concluded that the gendered patterns of caring in these groups included ethics of mutual support, solidarity and obligation. In fact, the parents interviewed did not link gendered patterns of caring to a devaluing of women's work. He concluded that the social dynamics and realities of ethnic minorities needed to be heard and that gender equality policies in host countries should be acknowledge by them.

Gender equality of the Nordic welfare model has been a widely debated issue, but the literature does not address the gendered aspects of the whole of society [7]. Dahlerup [8] and Borchorst [9] state that despite gender discrepancies of women in politics, leadership and unpaid domestic work, there is a strong belief that gender equality remains very much a reality in the Danish context. Borchorst and Siim [10] further argued that research had to "rethink the foundations of the welfare state and gender equality, from the perspective of migration and diversity". With the influx of refugees from different cultural backgrounds, cultural diversity has accentuated social and political inequalities in Nordic countries. More research needs to be done 'with those that the issue speaks for' as well as those 'who has the right' to define what women-friendly policies should be [10].

Pertinent to this discussion over the last two years, the world has witnessed a significant increase in forced displacement of millions of people due to conflict, extreme violence and climate change and forced migration has increasingly become one of the most significant global challenges of our time. At the end of 2018, there was a total of 25.9 million refugees globally, the highest number on record [11]. Migration is therefore one of the defining features of the 21st century and this is a major challenge to the achievement of the Sustainable Developments Goals for 2030. Consequently, a better understanding is needed of the relationships between migration and key development issues such as gender, education and labor. The relationship between migrants and the communities which includes both psychological and sociological processes of adaption, are key to a migrant's sense of belonging and adapting within their new host society [12]. The increasing presence of multi-ethnicity in Denmark also raises the important question of how Denmark's prominent culture of gender equality reacts to this modern phenomenon, particularly when integration is complex and touches frequently on the intersections of gender and ethnic diversity.

This paper is an attempt to engage with the problem described above. It describes a study examining the lived experiences and attitudes of a group of Eritrean parents living in Denmark. Its aim is to explore, through the voices of these parents, the growing tensions between Denmark's progressive gender norms, employment culture and gender equality policies, as encapsulated in the Nordic Welfare model as well as the cultural influences of forced migration. We attempted to describe the realities of how migration on the one hand creates very real opportunities for refugee women but also how gender norms are inherent in the societies they come from create vulnerabilities and can result in institutional failures. Whilst gender stereotypes are a common experience in the migration experience of migrant women, this paper focuses on women as mothers because as parents, they are liable to be

'left behind' in the progress towards achieving the 2030 sustainability goals [13]. There is a significant gap of knowledge in this area of research, especially regarding Eritrean parents.

*Gender Equality Under the Nordic Welfare Pillar*

Across the four Nordic states, Denmark, Finland, Norway and Sweden, family policies have been characterized by different dialectics and visions of gender equality [10]. Nordic countries encourage social equality and achieve high standards of living and actively promote gender equality policies. This approach enhances productivity and economic success [14,15]. Nevertheless, the Nordic welfare model, whilst being applauded globally, is facing a major challenge as a result of migration. Gender issues have been highly politicized and there have been strong debates about women's structural oppression, which may in part be attributed to the influence of Borchorst's [2] analysis of the gender system. Comparative gender research emphasizes that the Nordic welfare states share basic characteristics combining a large public sector with a dual breadwinner model. This leads to: (1) a family and welfare model where both partners are expected to be employed; (2) public welfare with extensive childcare services and generous parental leave schemes; (3) a relatively large number of women in senior political circles; and (4) gender equality as the accepted norm in public and private lives of citizens [15]. Paid parental leave and individual rights for fathers which include affordable public day-care for young children are claimed to be one of the hallmarks of the dual earner/dual carer model in the Nordic countries, where women and men equally share paid and unpaid work [14]. Despite this, inequalities still persist in all aspects of life such as women's weaker position in the labor market, gender pay gaps, an unequal division of domestic labor and care and an uneven distribution of assets and resources along gender lines [10,16]. Nordic countries have been generally treated as one coherent model because of their similarities. However, a closer analysis reveals considerable differences in the gender political models between these countries [17].

Refugee integration is a complex and multidimensional construct, incorporating the economic, educational, health and social context, and refers to the process of inclusion and participation, both economically and socially [18]. Its overall aim is that of the European Union's Sustainable Development Goal 16 which is to "promote peaceful and inclusive societies for sustainable development, provide access to justice for all, and build effective, accountable and inclusive institutions at all level" [19]. Promoting social integration is a new key concern, making migration and integration a challenge to the Nordic welfare model [15,20]. Importantly, integration is considered a two-way process and can only be successfully pursued by migrants when the host society is open and inclusive in its orientation towards cultural diversity [21]. Often the focus is on how refugees can or should adapt in order to integrate well. Successful integration, however, requires a social context with appropriate tailored policies which support inclusion and participation especially, especially in the context of the family unit, of women.

Bergeset & Ulvik [21] have argued that there are significant differences between male and female refugees in the integration process and this is why parenting and in particular motherhood is an important factor to consider when considering the experiences of refugee women who are attempting to integrate into Nordic countries such Denmark. Navigating a path forward in a new society includes multiple considerations. It is therefore imperative to determine how policies and services from an international, national, regional and local level either support or hinder integration of refugees by affecting the social context. Refugee settlement policies at both a national and local level influence refugee integration by shaping refugees' ability to participate socially and economically [22]. What it means to be a woman, a mother, a student, or a working adult can be very different in Denmark than in the countries where refugees come from. Notably, the presence of children tends to influence a women's desire and ability to work in general, and this is even more so in the case of refugee women. From a social constructivism standpoint, studies concerning refugee women's experiences and challenges of motherhood living under the Nordic welfare model is intrinsically connected to the issue of motherhood. There is a significant amount of literature related to the Danish approach to

motherhood but little if any documenting the experiences of refugee mothers. The family is considered as a migration unit and the gendered processes inherent in the cultural context of a migrating family can significantly affect that process.

## 2. The Danish Context

Denmark's once ethnically homogenous population has changed quite rapidly due to increasing immigration and this immigration has transformed the cultural face of the nation. Like other European countries, Denmark faces the challenge of culturally and economically incorporating an increasing number of foreign-born citizens. Integration is a complex notion, and this is especially true for immigrant women who occupy the intersections of gender and ethnic 'otherness' in Denmark. The increasing presence of foreign-born women in Denmark raises questions of how Denmark's prominent culture of gender equality develops and whether it affects these women. Research has highlighted the emergence of new inequalities among groups of women in the labor market, in the family and in this changing world [10]. This development has challenged the 'women-friendly' society 'where injustice on the basis of gender would be largely eliminated without an increase in other forms of inequality'. Siim [10] proposed that the new forms of inequalities among women can be interpreted as a Nordic Gender Equality Paradox between the relative inclusion of the native majority women in society and the current relative marginalization of women of diverse ethnic minorities on the labor market, in politics and society.

### 2.1. The Case of Eritrean Refugees in Denmark

Eritrean refugees constituted one quarter of all refugees to Europe in 2015 and were the most common migrants from Africa to the Central Mediterranean. In 2016, the Migration Policy Institute addressed the flow of Eritrean refugees from that country. In the previous year, the United Nations High Commissioner for Refugees [23] documented that at that time, 411,000 Eritreans were refugees, with 5000 people leaving Eritrea per month. Few countries in Africa or the Middle East offered these people either work or physical security. The European Union has however accepted Eritrean refugees in increasing numbers. In 2010 there were 4300 applicants and in 2015, the numbers had increased dramatically to 33,100 [24].

The issue of refuges has been strongly debated in the political arena in Denmark where increasing measures were put in place to limit immigration from Non-European countries. Compared to other countries, Denmark has the lowest rate of asylum seeker applications approved [25]. Since 1980, the figures for non-European immigration to Denmark has risen from 1% of the population to 8.5%. Political influences have played a role in migration, as there recently was a significant swing to the right in Danish politics. This change reflects directly onto immigration policies. The issue of refugees is currently strongly debated in the political arena. In 2018, the conservative reaction to immigration was severe and increasing measures were put into place to limit immigration from non-European countries. The transit from Eritrea, across the Mediterranean and finally to Denmark is daunting and perilous. Death, physical violence, sexual exploitation and forced return to Eritrea are only some of the dangers experienced almost universally by Eritrean refugees. Nevertheless, Denmark remains a desired destination because of its social security, good education and the welfare state providing a secure safety net including housing and social services.

### 2.2. Gap in Research Relating to Eritrean Refugees in Denmark

Minimal research has been directed towards the lived experience of Eritrean parents in Denmark which are attempting to cope with the double-integration—firstly their own integration into Danish society and then the integration at another level by their children. In 2018, in a review addressing child refugees, only two of the 46 articles included in the review specifically named Eritrea as the country of origin [26]. This is surprising given the comparative volume of migration from Eritrea compared to other countries, or even other countries in the Horn of Africa, that this specific linguistically and

culturally unique refugee population has not been the subject of more in-depth study, particularly relating to parenting. The objective of the study was to address this gap. Through the medium of narrative [27], we sought to elucidate on the lived experience of Eritrean refugee parents already residing in Denmark. A systematic literature review confirmed that there were no studies conducted regarding the lived experience of Eritrean refugee parents. During the time of the review, a study conducted in Norway addressed how refugee and immigrant mothers experienced 'doing parenting' in a Norwegian social context [28]. This study indicated that the intercultural contact between refugees and the Norwegian's society was filled with difficulties and ambiguities' [29]. The lack of data in the Nordic countries has been a strong driver for this study.

## 3. Aims of the Study

A significant, unresolved practice research issue in Aalborg Kommune exists [30]. This relates to the limited knowledge of the challenges, which are experienced by Eritrean families in transitioning from their homeland to Denmark. To understand the parenting experiences of Eritrean refugee parents in Aalborg, a study was conducted in 2019. The aim of the study was to explore the lived experience, perspectives, practices, and challenges of Eritrean parents in the Aalborg Kommune. The diversity of Eritrean parents' lived experience in raising their children in a host country setting with very different social and values context is an issue, which is of topical and current significance [20]. We were of the opinion that the narratives of the lived experiences of Eritrean parents would help us to understand how they make meaning of these experiences, and that this information will inform future social services that are positively delivered to this vulnerable community.

### 3.1. Theoretical Underpinning

The theoretical framework for the study had its basis in both a phenomenological perspective and narrative theory. The phenomenological assumption in narrative analysis is that by lived experience, the meaning of which is ascribed to a phenomenon [31]. By narrating their experience, a person has the opportunity to negotiate its meaning, evaluate its significance and then assess the next step of action [32]. Analytically, the aim in this study was to distinguish between the phenomena that a person encounters in life and the way the person perceives it as providing meaning to events. Berry's [20] conceptual framework of immigrants' acculturation to the host was useful when seeking to understand an immigrant's adaption to a new society.

### 3.2. Methodology

This qualitative study was small, with six Eritrean parents recruited by means of a snowball sampling. A purposive approach to recruitment was undertaken in order to comply with the need for homogeneity of experience in the sample [33]. Six Eritrean refugee parents were interviewed, and a description of the study participants is provided (Table 1). This was sufficient to enable the researcher to evaluate the lived experiences of these subjects, focusing more on the particular aspects of these experiences. A participatory approach, characteristic of Practice Research, was adopted for this study and whilst there are different definitions of what makes research participatory, the central aim regarding this approach was to position those being 'researched' as active partners, creating a reciprocal relationship between the researcher and those that were researched. This approach allowed for mutual exchange of information, giving a voice to individuals and communities [21]. The impact of gender equality policy and the welfare state on the family lives of refugees can only be understood from the lived experience of families. In preparation for the study, the researcher had several meetings with the Eritrean community representatives, where the study was introduced and explained.

**Table 1.** Characteristics of Participants.

| Country of Origin Eritrea | 6 |
|---|---|
| **Age Range** | |
| 20–30 years | 4 |
| 30> | 2 |
| **Length of time in Denmark** | |
| 1–5 years | 2 |
| 5–10 years | 4 |
| **Living Status** | |
| Living alone | |
| Living with children (under 5 yr old) and husband | 6 |
| **Religion** | |
| Orthodox Christian | 6 |

A focus group with community members was developed where members of the community were asked to document what they were comfortable with regarding the questions. The co-creation of the questions enabled discussions to take place, deepening the exploration of topics raised in the interview questions. Following the focus group and the several discussions with the community leaders, the interview questions were finally developed in consultation with the Eritrean interpreter, who serves as an important source of cultural information. This approach was grounded in the African tradition of communicating events orally through story telling. Harrell-Bond and Voutira [33] state that storytelling allows refugees to make sense of traumatic experiences and this approach also provides an opportunity for participants to share knowledge readily and recount family experiences as part of the narrative. As well as providing verbal and written information detailing the research, the researcher was invited to participate in several unique Eritrean cultural ceremonies, such as the "smoking ceremony". These ceremonies allowed the researcher to gain a deeper understanding of the Eritrean culture and enabled her to connect with respect and in a culturally appropriate way with the participants in the study. Whilst this study involved a broad investigation of the family life of Eritrean refugees, this article will focus on the findings in relation to the Eritrean mothers and the impact of Danish gender equality policies as they related to them.

*3.3. Results*

We applied the Bakhitinian concept of multi-voicedness to explore an organic approach [34] and three themes were found relating to gender and the role of women in Eritrean society. These themes are illustrative of the tensions that refugee status brings to bear on the role of men and women in the family unit, education and employment.

*3.4. Tension between Integration Policy and Culture*

The interviewees experienced many differences that were categorized as (1) societal, (2) language, (3) cultural and (4) educational differences. These differences indirectly influence parenting experiences and affected the parents' overall wellbeing. The fears and uncertainties were projected into the space of parenting, particularly related to the needs that could arise, in the future, for their child or children. Initially, the parents faced several societal differences. Arrangements on how to provide for basic needs for their families such as housing, transportation and paperwork were foreign to them. Moreover, there were different laws, rules, and different social mores such as needing to maintain communication with the Kommune and leaving their children in day care while they worked. The example below shows the importance of familiarizing themselves with the society and gaining knowledge to provide for the children but also highlights the significant difficulties, fears and expectations these refugees encountered as mothers in Denmark.

> *MO 1-It is important that I teach my children both cultures from an early age but very important at the age of 13 years old. I don't want them to experience culture conflict and rebel.*

It was also considered important for the mothers to rapidly acquire the knowledge of how to be able to teach their children how to function in the new society by attending social outings such as mother's group activities. This had a direct effect on their parenting practice, including by reducing the social isolation that they felt. The mothers stated that a lack of knowledge affected the ability to predict what their children would encounter in the future and in that, an essential parenting practice had been lost by the parents. They themselves felt unable to be able to provide their children with knowledge about how to operate in Danish society.

Being separated from their family created a shift from the collective nature of their households in Eritrea to a more individualistic household and these findings reflect how the composition of families can influence changes in parenting experiences [35]. The mothers described periods of intense loneliness frequently as a result of the changed responsibilities for the mother. The changed responsibilities such as looking after their children at home also prevented them from attending Danish language classes and a circle of negative issues emerged from this.

However, the mothers spoke of the opportunities in Denmark and their appreciation that they were in a free society, without the constraints they had experienced in Eritrea.

> *MO 2-Here in Denmark, they take care of the rules and Denmark has a good system, you have no conflict with other people. You should take care of yourself and not worry about other problems. If you have a problem, Denmark has a good way to solve it and has a good system to solve it. To be here in Denmark is very nice there is no conflict with others, I like the multicultural. Everyone has their own ways. If you make a problem, you get a lot of problems.*

*3.5. Conflicting Expectations between Traditional Parenting Roles and the Expectations for Women to Work in the Welfare State*

Employment in Denmark is strongly predicated on the ability to speak Danish. The enrolment in Danish language classes is a prerequisite for obtaining social security monthly payment. This language education is therefore an integral part of the process of employment and enhancing earning power, as well as a Danish Integration policy requirement for newly arrived immigrants. Addressing the question of how they perceived their mothering experience with gaining employment and further education, all the mothers expressed that they would like to have a job. The women offered a diverse range of sentiments about motherhood and working which showed that motherhood experiences played a large role in these women's attitudes about working. One mother explained her frustration about the fact that the necessity of work made it difficult for them to spend time with their children. She was not enthusiastic about the structural support such as daycare options and was unhappy about leaving her child in daycare in order to work. The women expressed several different reasons for being reluctant yet willing to work.

One mother articulated this sentiment when she said:

> *MO 1-Here, if you do not work or go to school, you cannot make money. So you have to! My children have been in the day care and there is not enough time for me to be with them. That is a problem for me.*

All the mothers accepted that they would have to rebuild their lives and showed flexibility and resilience in how they would do this, as can be seen below.

The Eritrean refugees had lost their country and settled in a country with a dramatically different culture and worldview. Their impression was that Danes did not have an understanding of them as people or the experiences which they had been through both in Eritrea and in the trajectory of migration to Denmark. This loss was characterized by all the participants as a loss of stability currently—even compared to the fragile sense of stability in Eritrea, as demonstrated below;

*MO 1-The idea of being something in your country and coming here as a refugee. It's a very big difference.*

*MO 1-To start from the beginning was very hard . . . not knowing the language or able to provide for my family . . . I was scared I couldn't get a job.*

These quotes demonstrate an existential change, have lost not only their stability, and their status in the community but also income and ability to anticipate a future at all. For all the interviewees, the emphasis suggested that this was present in the very beginning of the time in Denmark. It also shows a drift towards the concept of "them" and "us" was developing.

One mother accepted that she would have to rebuild her life, showing resilience and flexibility in how she was attending school so that she could get a job working in an aged care home.

*MO 1-I am a qualified civil engineer, however it is too hard for me to validate my certificate here. So, I have completed an aged care course which is completely different to my background. But it supports my family.*

Even though she liked to study and believed it was her responsibility to contribute to the household income, she was unhappy with the idea of working because she thought her children, who were three years old and one years old, were still too young. She explained a practical reason why children's age should determine when a mother works:

*MO 2-I think mothers who have children must stay at home until the children grow up, and then they can work. It is like that back home. I think when my children can eat, dress themselves, speak, and understand, I can then go to work. Before that, it is impossible for children to be by themselves.*

However, one mother stated that she enjoyed the social experience of attending the local Mother's group provided by the municipality. She had stated in previous discussions that she found it hard coming from a large family to living with just her daughter and husband.

*MO 3-Once a week, I attend a play group run by the Kommune with my daughter for other refugee mothers and it is nice to talk with them. It helps reduce my loneliness sometimes. It does not matter that it's not all Eritrean mothers.*

This statement shows how attending a mother's group decrease the loneliness she felt, hence reducing social isolation.

Two mothers provided an emotional explanation regarding their mothering responsibilities.

*Mo 3-First I must take care of my family. I must do what they need. When they are young, they need me. When they are older, I look for my future.*

*MO 1-I must at home all the time. My job is taking of my children. My husband goes to work. Being a mom is about taking care of your children and the house; it's a very good job.*

These mothers clearly wanted to work but were reluctant to do so due to their values and the importance to themselves of being a good mother. They clearly stated that they would feel more comfortable working when their children were older but due to the strict integration policies, they needed to participate in an education program or work in order to get their monthly allowance from the government.

*3.6. Feelings of Loss of Multi-Layered Family Structure and Consequent Isolation*

What was present in all the stories was how much the women missed their family. For one mother, this resulted in intense loneliness and a sense of isolation.

> *MO 2-In Eritrea, family is very important. We have a big family. We support each other. Here I feel alone. I only have my husband.*

Her support-system had been reduced from a network of family to one person. The change had little to do with exposure to, or comparison with the Danish culture. It was experiential—a simple loss of community and within that community, family connections. Separated from their family created a shift in family dynamics from the collective nature of households in Eritrea to a more individualistic household [36]. The changed responsibilities including looking after their children at home prevented the women from attending Danish language classes and a circle of negative issues emerged from this.

## 4. Discussions

This study was essentially exploratory in nature and suggested that Eritrean mothers' main concern is for their children and the retention of what they view as defining aspects of being Eritrean. Their opinion highlights the cultural gap between the worldview exemplified by the Nordic Welfare Model and the Eritrean culture. Since the late 1990s, Nordic countries have become more ethnically diversified. During this process, Gender Equality in relation to multicultural minorities has been the subject of contentious political and national debate. In the early 2000s, many authorities questioned whether the ideal of woman-friendly policies is based on the livings conditions of white middle class women and whether it in fact excludes minority women [37]. It has been asked whether minority women have gained a voice in the same way as Nordic women did in the 1970s [38,39].

The dominant emerging themes related to strategies used by the mothers to bridge the two cultures while retaining their ties to the local Eritrean community. All the mothers felt that their support-system had been reduced from a network of several families to often only their nuclear family and some friends. This signified a loss of community exemplified by the loss of extended family and a circle of friends. There was no desire to participate in the equal-gender economic view, which is a characteristic of the Nordic Welfare model. This study emphasizes the importance of mothering and family life for Eritrean refugee mothers.

Boswell and Geddes [39] argued that it is important to understand the link between the family as a migration unit and gendered processes and how this enhances understanding both of migration and of political and social processes associated with it. The family unit is significant in Danish policies, which are aimed at promoting immigrant integration work through family structures. Denmark identifies families as an important source of social values and targets them as key actors in the integration process. Our findings indicate that there is a cultural divide between Nordic and the Eritrean cultural values. As a result, mothers described periods of increasing loneliness because of the changed responsibilities inherent in being a mother in a foreign society without the support of the extended family.

Unexpectedly, because of the preparatory and final discussions held with Eritrean leaders throughout the study, community-based interventions were recommended with education and employment considered to be advantageous. Studies in Australia have shown that community-based interventions are effective in facilitating the inclusion and social participation of refugees [40]. Thematic analysis of the group dynamics suggest that a community/group approach would be more appropriate for the Eritrean mothers than a one-on-one approach and social work interventions should construct interventions around a community or group model.

## 5. Conclusions

Despite Denmark being successful in implementing a successful dual breadwinner model it is apparent from the analysis of Eritrean women narratives, this vision does not eliminate gender differences in the context of a rapidly changing multi-ethnic and multicultural society. In this changing world, this study has highlighted the emergence of new inequalities among minority women in the labor market, family and gender equality policies. Key aspects of the Eritrean families' resilience emerging

from this study was the maintenance of language and religion and the centrality of their children in their decision making. The findings also suggest that Eritrean motherhood is influenced by external factors such as interactions with Danish institutions and their new social environment. From the interviews, the mothers described many different strategies of how they passed on their mother tongue, traditions, and values to their children to ensure they maintained their culture and connections in the Eritrean community. The results suggest that further research needs to be implemented regarding motherhood and the social integration of refugee families in Europe. Motherhood is an important factor in studies of refuge women integrating into Nordic countries such Denmark. From a social constructivism standpoint, the study of motherhood is intrinsically connected to the study of womanhood, and qualitative studies concerning refugee women should account for the experiences and challenges of motherhood living under the Nordic welfare model. Although there is a wealth of literature on Danish constructions of motherhood, little if any of it considers the experiences of non-native, migrant mothers living in Denmark. These narratives of the Eritrean refugee experiences have provided us with an understanding of how Eritrean mothers creating meaning of their lived experiences of migration relating to their children. Our study indicates that a narrative approach serves as a useful construct for better understanding and the subsequent creation of a platform for collaboration [21].

Despite the challenges faced by Nordic countries due to the increasing multi-ethnicity as a result of population-shifts, the Nordic welfare model has important strengths and is equipped to deal with these challenges when it comes to promoting integration including the continued social investment in integration programs for migrants, as well as the universal and generally transparent benefit system. With these changes with challenges such as migration and different forms of family structures, gender equality policies will have to be adapted to suit the needs of the diverse populations that are developing in the Nordic countries to offset any particular challenges immigration poses for the Nordic welfare states. This realization indicates the direction of future research which will inform and support successful integration of refugees into Nordic countries.

**Author Contributions:** Conceptualization, N.J., J.C.A. and T.M.-O.; methodology, N.J., J.C.A. and T.M.-O.; supervision, J.C.A. and T.M.-O. All authors have read and agreed to the published version of the manuscript.

**Funding:** This research received no external funding

**Conflicts of Interest:** The authors declare no conflict of interest

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
