# Peer review of "Migration as a Challenge to the Sustainability of Nordic Gender Equality Policies as Highlighted through the Lived Experiences of Eritrean Mothers Living in Denmark"

_sustainability, doi:10.3390/su122310072_

Round 1

Reviewer 1 Report

Thank you for giving me the opportunity to review the manuscript Migration as a challenge to the sustainability of Nordic gender equality policies as highlighted through the lived experiences of Eritrean mothers living in Denmark. The manuscript applies a narrative approach, which is based on interviews with migrant mothers residing in Denmark. With a particular focus on the lived experiences and attitudes of migrant mothers living in Denmark, particularly how they made meaning of their lived experiences of migration relating to their children, the paper explores the growing “tensions between Denmark’s progressive gender norms, employment culture and gender equality policies” and “the cultural influences of forced migration.” It shows that the realities of these women, as well as the existence of gender norms, create both opportunities and obstacles for them in integrating on the labour market. It concludes that motherhood is a key factor in studies of refugee women integration into Nordic countries such as Denmark.

The paper should be commended for addressing an important topic, which has been quite under-researched so far. Having this said, it needs some more polishing before it is ready for publication.

The following major and minor comments and queries should be addressed in the revision of the manuscript:

Major comments:

  1. The manuscript needs to problematize the status of gender equality in Denmark and in the Nordics. For instance, the statement that gender equality has already been achieved for the majority of Danes (p. 1, row 24), is highly contested in research.
  2. In a similar way, the conceptualizations of the Nordic countries as similar as pertains to gender equality (as well as migration) need to be problematized. In fact, recent research on gender equality suggests that there are apparent differences between the Nordic countries. Hence, the paper needs to bring out the complexities with regard to similarities and differences of the Nordic countries.
  3. Likewise, the literature on gender equality in Denmark and in the Nordic countries is quite old. More recent scholarship should be included, for instance Anette Borchorst and Drude Dahlerup’s edited book on gender equality in Denmark.
  4. More information on the interviews and interviewees are needed. What criteria for selection of interviewees were used? How were interviews conducted? Age of interviewees, number of children, etc. More quotes are also needed to substantiate the conclusions.
  5. In the concluding part, new information and conclusions that have not been analyzed in the text are introduced. For instance, aspects such as the maintenance of language and religion, and the influence of external factors, such as interaction with Danish institutions, on Eritrean women’s motherhood, are presented. It is also mentioned that women pass on their mother tongue, traditions and values to their children. These parts should either be removed or presented in the analysis.
  6. The text on recommendations for service providers and future reform of Nordic gender equality policies should be removed. The paper does not address this topic in detail. I suggest that the author stresses the need for more research in the field.

Minor comments:

  1. Abstract and introduction of paper: the papers main research question as well as major conclusion should be spelled out clearly and placed up-front.
  2. Page 3, row 131. What does “obtain monetary welfare assistance” mean in this context? Isn’t women’s engagement in employment and education about “achieving empowerment and economic independence?”
  3. Page 4, row 186-188. What does a community-based approach mean? Please clarify.

I wish the authors good luck in revising the text.

Reviewer 2 Report

GENERAL REMARKS

For the authors’ guidance, my evaluation and some constructive remarks that would help to improve the paper’s quality are included below:

(1) I appreciate there has been a lot of reading and ground covered. The paper would benefit from some minor revisions. Why the authors did not specify particular research inquiries that are relevant to the study? Likewise, adding some follow up questions can be beneficial. In case when research questions are identified, then readers can easily conceive the framework of investigation at first glance. They understand somehow “what” the paper attempts to do and they get the “how.” However, the most interesting thing, the “why!!!” ought to be systematically conceived very well. I recommend the authors to come up with clarifications of why the research is interesting and relevant to the field.

(2) I believe the authors will be more successful if they focus on clear, direct arguments that build up one at a time and add up sequentially to persuasive and tightly focused cases.  My own struggles with the task of writing scientific articles for peer-reviewed journals suggest that it is worth spending a long time in the detailed planning of papers around a sequence of bullet-point arguments. Plans are best if argument-driven.  It is worth resisting writing full prose until a very detailed plan has been developed that seems to work, and then supporting material can be added after each bullet point argument. Certainly, I believe the authors are investigating interesting research areas where there is good scope for innovative new contributions and publications.

(3) I think the authors have successfully linked up some perspectives and concepts of various disciplines. However, this nexus and interrelationships between different disciplines ought to be reinforced, systematically. I recommend quoting the following reference source: de Haas, H., Miller, M. J., & Castles, S. (2020). The Age of Migration: International Population Movements in the Modern World. Red Globe Press.

(4) The methodology of the paper seems a bit vague. There are little substantial clarifications. The paper needs much more work in relation to its structure, methodology, objectives, and discussion. Moreover, the authors may enhance the methodology of their research by citing the publications of E.G. Guba, Y.S. Lincoln, N.K. Denzin and so forth.

(5) I recommend the authors to clarify their methodology in detail, making sure that their planned methods/research tools are fully detailed (e.g., the narrative approach). They ought to give attention to justifying their chosen methodology in terms of demonstrating applicability, adjustment, and usefulness in the paper.

(6) All in all, I recommend the author(s) to reconsider the approach adopted here; think about the main empirical question they wish to examine; make sure the literature review is a lot more cohesive, and make sure the link between the research question and empirical results is a lot 'tighter' than presented herewith.

(7) MDPI - Sustainability is a remarkable peer-reviewed journal. A referee ought to recommend a manuscript that is deemed a great contribution to the journal’s future achievements. Consequently, the paper demonstrates rigorous research outcomes/findings that can be useful for the readers of the MDPI - Sustainability.

SPECIFIC REMARKS

(1) The authors started with a quotation: “Immigrant women do not have contact with the surrounding society. They do not know their rights, and they do not determine their own existence. The lack of gender equality for many immigrant women is part of the new inequality. It is important to create equal opportunities between men and women because this contributes to creating strong cohesion in society (pp.1).” A Danish parliamentarian was quoted. However, social scientists are perfectly conscious of the fact that gender equality, fairness, anti-discrimination, justice, social cohesion, and many other related concepts should initially be argued in the context of EU law. EU political issues come later.

(2) I think “ethnic otherness” has been instrumentalised as an abusive notion. I recommend Steven Vertovec’s “ethnic diversity” and “third-country nationals (TCNs).” A cosmopolitical perspective towards the multicultural minorities will be fine.

(3) What differs the Nordic welfare model from other models? Why is it so unique? Please, clarify these by considering the “three worlds of welfare capitalism (Esping-Andersen, 1990; Emmenegger, Kvist, Marx & Petersen, 2015).” You may also cite Esping-Andersen (2009) Incomplete Revolution: Adapting Welfare States to Women’s New Roles. Cambridge: Polity Press.

(4) There is an ambiguity in the nature of the refugees' integration process. One needs to argue the “shared” responsibilities of the integration process. As you know many internal and external stakeholders are involved in this process. What is the role of civil/civic society organisations for the integration of refugee women in Denmark?

(5) The authors did not classify their components and determinants that are beyond migration and sustainability nexus. There are many interrelated impact factors. They ought to argue stakeholders in detail. They can classify stakeholder groups, types, and parts. There are public stakeholders, private stakeholders, civil society stakeholders, and so on. The clarity of the usage of stakeholder theory is a bit vague.

(6) Is social constructivism the research paradigm that the authors intended to apply? If the answer is yes, then they ought to clarify the ontological, epistemological, hermeneutic, and phenomenological presumptions of social constructivism in a paradigmatic manner.

(7) Cultural, ideological and political biases ought to be observed and analysed in a more critical way. More critiques towards globalisation, (post)modernism, consumerism, marketisation, welfare capitalism, imbalances of power and societal relations, legitimacy, and so on can make sense. Ways of knowing in participatory research are quite significant for critical hermeneutic subjectivity. In this context, the principle of inter-subjectivity and “verstehen” concept of Heidegger are also associated with a transcendental and pragmatic sense of sustainability, migration, family policies, and so forth.

Round 2

Reviewer 1 Report

Dear Author, 

The paper has been substantially improved. I have just identified a few minor aspects that need to be attended to.

  1. Row 354. A word is missing (five themes are listed, but theme 5 is missing)
  2. Interviews: Information on when (year) the interviews were conducted is missing.
  3. Interviews: information on interviewees is missing, for instance in terms of age, number of children, years in Denmark.
  4. Row 405-406. "pressure for women to engage in employment education so that they would receive their social security each month". What does "employment education" mean? Is this a special measure offered to migrants/unemployed people by the Danish Agency for Employment in order for migrants/unemployed people to become prepared for paid employment? 
  5. Good luck revising the text! 

Author Response

Dear reviewer 1,

Thank you very much for reviewing the article.

I appreciate very much the constructive feedback you have provided.

Please find below the recommendations that you have made and my comments in how I have responded to them.  

Kind regards,

Natalie Joubert

1.          Row 354. A word is missing (five themes are listed, but theme 5 is missing)

I have deleted the letter 5

2.          Interviews: Information on when (year) the interviews were conducted is missing.

I have included the year 2019

3.          Interviews: information on interviewees is missing, for instance in terms of age, number of children, years in Denmark.

I have inserted a table detailing the characteristics of the parents interviewed

4.          Row 405-406. "pressure for women to engage in employment education so that they would receive their social security each month". What does "employment education" mean? Is this a special measure offered to migrants/unemployed people by the Danish Agency for Employment in order for migrants/unemployed people to become prepared for paid employment? 

I have rewritten the sentence so that it is more clear as to why this is important